# Advances in siRNA Drug Delivery Strategies for Targeted TNBC Therapy

**DOI:** 10.3390/bioengineering11080830

**Published:** 2024-08-14

**Authors:** Md Abdus Subhan, Vladimir P. Torchilin

**Affiliations:** 1Division of Nephrology, University of Rochester, 601 Elmwood Ave, Rochester, NY 14642, USA; 2Department of Chemistry, Shahjalal University of Science and Technology, Sylhet 3114, Bangladesh; 3Center for Pharmaceutical Biotechnology and Nanomedicine, Department of Pharmaceutical Sciences, Northeastern University, Boston, MA 02115, USA; 4Department of Chemical Engineering, Northeastern University, Boston, MA 02115, USA

**Keywords:** triple-negative breast cancer, siRNA, biomaterials, nanomaterials, targeted therapy

## Abstract

Among breast cancers, triple-negative breast cancer (TNBC) has been recognized as the most aggressive type with a poor prognosis and low survival rate. Targeted therapy for TNBC is challenging because it lacks estrogen receptor (ER), progesterone receptor (PR), and human epidermal growth factor receptor 2 (HER2). Chemotherapy, radiation therapy, and surgery are the common therapies for TNBC. Although TNBC is prone to chemotherapy, drug resistance and recurrence are commonly associated with treatment failure. Combination therapy approaches using chemotherapy, mAbs, ADC, and antibody–siRNA conjugates may be effective in TNBC. Recent advances with siRNA-based therapy approaches are promising for TNBC therapy with better prognosis and reduced mortality. This review discusses advances in nanomaterial- and nanobiomaterial-based siRNA delivery platforms for TNBC therapy exploring targeted therapy approaches for major genes, proteins, and TFs upregulated in TNBC tumors, which engage in molecular pathways associated with low TNBC prognosis. Bioengineered siRNA drugs targeting one or several genes simultaneously can downregulate desired genes, significantly reducing disease progression.

## 1. Introduction

Cancer is the second leading cause of mortality after cardiovascular disease, causing the deaths of 602,347 people in 2020 in the USA [1]. In 2022, around 20 million new cancer cases were diagnosed, and 9.7 million cancer patients died worldwide [2]. Cancer cases are predicted to increase to 35 million by 2050 [2]. Breast cancer is the major reason for mortality among women. The most aggressive, highly heterogeneous subtype of breast cancer is TNBC, which lacks expression of ER, PR, and HER2 markers. About 15 to 20% of all breast cancers are recognized as TNBC [3,4]. Currently, effective TNBC therapy is limited, and chemotherapy is the only good option. Based on the overexpression of ER, PR, and HER2, breast cancer patients are offered targeted therapy. In the absence of such markers, therapy for TNBC is more difficult to determine, leading to a higher rate of death among women worldwide. TNBC is highly proliferative and rapidly metastasizes to the brain, bone marrow, lung, and liver. TNBC affects younger women under 40 years of age [5]. In the clinic, TNBC is treated with chemotherapy, which leads to a poor therapeutic outcome. Extensive investigations using clinical, histological, and molecular profiling demonstrated the heterogeneity of TNBC tumors. Among the four subtypes of TNBC, women with the luminal (20%) or basoluminal (28%) subtypes exhibited a significantly worse rate of survival than patients with the basal A (26%) or basal B (26%) subtypes of TNBC [6].

The initial treatment of TNBC is surgery and radiation therapy followed by chemotherapy using a conventional method. Common chemotherapeutics used in TNBC treatment include taxanes, anthracyclines, paclitaxel, and cisplatin/carboplatin drugs. Chemotherapy can be applied by two different approaches: neoadjuvant chemotherapy (NACT) or adjuvant chemotherapy (ACT). Since TNBC lacks suitable markers, a targeted therapy for TNBC patients is a challenge. In recent years, targeted therapies of TNBC have involved innovative treatments such as PARP inhibitors, different signaling inhibitors, RNase inhibitors, antibody-drug conjugates (ADC), and immune checkpoint inhibitors (Figure 1) [4]. Additionally, biomaterials that target signal transduction, angiogenesis, epigenetic modifications, cell cycle, and EMT (epithelial-mesenchymal transition) are undergoing preclinical and clinical investigations for TNBC therapy [7,8,9].

Novel nanobiomaterials have a variety of excellent physical and chemical properties such as size, shape, surface area, and surface chemistry. Nanobiomaterials possess important biological functions including targeting specificity, contrast agents, and drug delivery vectors [10]. Major types of nanobiomaterials are organic and bio-based materials. Organic biomaterials can be fabricated from peptides, proteins, DNA, RNA, viruses, and other biopolymers, and are suitable for their applications as biosensors, biomimetic mineralization, bioimaging, and drug delivery [11]. The US-FDA-approved organic polymer materials including polyethylene glycol (PEG) and poly-lactic-co-glycolic acid (PLGA), have been utilized extensively for drug delivery, particularly in cancer therapy [12,13,14]. These nanobiomaterials are appropriate for use as carriers for siRNA therapeutics.

The bio-based biomaterials originate exclusively from cells, bacteria, and viruses including exosomes, protein-based nanosystems, and virus-like NPs (VLPs) [15,16,17,18,19]. Nanobiomaterials can be classified based on their origin as natural or synthetic. Natural biodegradable polymer materials suitable for drug delivery systems include proteins (collagen, albumin gelatin, etc.) and polysaccharides (starch, dextran, chitosan, etc.). Natural biomaterials are more attractive as host tissues easily metabolize them. Conversely, synthetic nanobiomaterials are appealing due to facile synthesis and tunable properties [20,21].

Furthermore, biomineral-based inorganic nanobiomaterials including calcium phosphate, calcium carbonate, and calcium silicate are the chief inorganic ingredients of biological hard tissues. They take part in the normal metabolism of the living body. Therefore, such inorganic nanobiomaterials are biodegradable nanocarriers suitable for drug delivery. They are auspicious for applications in the clinic. Inorganic-based nanobiomaterials are significant for drug delivery due to their easy fabrication and structural rigidity [22].

RNAi (RNA interference) is a promising approach for TNBC therapy. RNAi can be achieved by using miRNA, shRNA, and siRNA. RNAi facilitates the targeting and inhibition of specific genes, proteins, or transcription factors (TFs) associated with growth factors, signaling molecules, and controllers of apoptosis. Selective targeting of these molecular factors by siRNA-based therapies demonstrated potential for inhibiting specific cancer genes including those associated with TNBC tumor development and progression with minimum side effects [23,24]. Effective systemic siRNA delivery to target cells required overcoming several extra- and intracellular hurdles. Proper utilization of suitable nanomaterials or nanobiomaterials can effectively encapsulate siRNA and deliver it to target tumor cells, overcoming physiological delivery barriers, and silence specific target genes to inhibit tumor cells without side effects. The application of siRNA therapies in TNBC can offer the advantages of multiple gene suppression, mitigating drug resistance that may ensue due to repeated drug administrations. This multi-faceted strategy is effective and prospective to improve therapy outcomes of aggressive TNBC tumors in patients [24].

Here, we discussed major problems represented by TNBC. Given the difficulties in responding to classic treatments, we presented novel therapeutic possibilities for the treatment of aggressive TNBC. Subsequently, we analyzed and presented biomarkers, genes, TILs, and their role in TNBC. We explored current therapeutic strategies, and finally focused on nanomaterial- and nanobiomaterial-based siRNA drug delivery systems for TNBC therapy exploring effective targeted therapy approaches to combat TNBC.

## 2. Significant Roles of Biomarkers, TILs, and Genes in TNBC

The immune response can inhibit tumor cell activity without affecting healthy cells in a significant and specific manner. Tumor growth can be controlled by cytotoxic innate and adaptive immune cells. The immune system surveys the human body for tumors to recognize and eliminate them based on the antigens expressed by tumor cells [25,26]. The immune system can either inhibit tumor cells or offer a favorable tumor microenvironment (TME) for tumor development depending on the disease stage. As tumors consist of different cell types, interactions of these cells within the TME control the tumor growth process, and the immune cells in the TME can efficiently enhance or reduce tumor growth [3].

Body immune systems work to inhibit cancer; however, evasion of the immune response can result via several mechanisms. These include PD-L1 protein expression on tumor cells. Binding to PD-1 or B7, PD-L1 can exert immune suppression [27]. B7 is a type of integral membrane protein found on the activated antigen-presenting cell (APC) surface when paired with either CD28 or CD152 (CTLA-4 (cytotoxic T-lymphocyte-associated protein 4)). There are two major types of B7 proteins, B7-1 or CD80, and B7-2 or CD86. CD80 is found on dendritic cells, macrophages, and activated B cells, and CD86 on B cells. PD-L1 expression has been detected in TNBC, but not in other breast cancer cells. Expression of PD-L1 in tumors and the presence of tumor-infiltrating lymphocytes (TILs) with PD-L1 are allied histologically to a high degree and with more lymphocyte infiltration of tumors [28]. PD-L1 expression is linked with the CTLA-4, IDO1 (indoleamine 2,3-dioxygenase 1) expressions, and mutations of the *BRCA1* gene [29]. Treatment with immune checkpoint inhibitors (ICIs) in combination with chemotherapeutics can augment the prospect of TNBC therapy by enhancing the tumor-associated immune response in breast tumor cells because chemotherapy can enhance the release of antigens, tempt the expression of major histocompatibility complex (MHC) class I molecules, neoantigens, and PD-L1, and induce dendritic cell stimulation. Thus, chemotherapy can improve the immune response following or during ICI therapy. Many investigations are ongoing utilizing PD-1 and PD-L1 inhibitors combined with other checkpoint inhibitors, chemotherapy, and radiation therapy. These approaches may improve immune-checkpoint-inhibitor-induced immunotherapy for TNBC in clinics [30].

The activation of T cells is enhanced by the binding of B7 molecules to CD28 and subdued by binding to CTLA4. The role of TILs is impeded by the activation of the CTLA-4 receptor [31]. TILs can induce robust tumor regression in cancer patients [31]. CTLA-4 overexpression has been detected in TNBC [32]. CTLA-4 overexpression is linked with mutated BRCA1 TNBC [33]. Many investigations with CTLA-4 inhibitors such as ipilimumab and ipilimumab plus nivolumab in comparison with neoadjuvant chemotherapy, the combination of nivolumab with NACT, in patients with primary TNBC disease are now in pre-clinical and clinical stages [34].

TNBC patients demonstrated common founder mutations in the highly penetrant cancer susceptibility genes BRCA1/2. Studies of TNBC patients have revealed an association of TNBC with mutations in several moderate penetrant breast cancer susceptibility genes, including *ATM*, *BARD1*, *BARIP1*, *NBN*, *PTEN*, *RAD51C*, and *RAD51D* (Table 1) [35]. Mutation in *BRCA1/2* tumor suppressor genes that encode proteins associated with DNA double-strand break repair occurs in 10–35% of TNBC. Mutation occurs in germline and somatic cell lines [36,37]. PARP members are involved in the repair of DNA single-strand breaks. TNBCs (15–20%) have germline *BRCA1/2* (*gBRCA*) mutations [38]. Tumors that share molecular landscapes of BRCA-mutant tumors, that is, those with BRCAness, may respond to similar therapeutic strategies [39]. *gBRCA* mutation and BRCAness grades are linked with enhanced sensitivity to chemotherapy and improved clinical results [38,40]. TNBC cells with a *gBRCA* mutation are prone to PARP inhibitors due to the synthetic lethality mechanism, causing failure of DNA repair [41]. The *gBRCA* mutation is allied with an enhanced PCR rate and prognosis in patients with TNBC.

The existence of TILs is the most reliable prognostic factor in patients with TNBC [42]. TNBCs possess a high immunogenic character as mirrored by stromal TILs, and TILs have a predictive and prognostic role in TNBC patients [43]. IDO1 is another enzyme associated with immune regulation in TNBC. IDO1 is formed primarily by stimulated macrophages, and overexpression is correlated with poor prognosis in breast cancer patients [44]. Some of the prognostic and predictive biomarkers in TNBC are summarized in Table 2 [45].

## 3. Currently Utilized Therapeutic Targets in TNBC

Vascular endothelial growth factor (VEGF) binds to its receptor and enhances the growth and expanse of TNBC tumors. Patients with TNBC demonstrate a high level of VEGF expression [3]. Tyrosine kinase inhibitors with chemotherapeutic drugs are effective against TNBC with VEGF expression. Anti-angiogenic drugs such as apatinib and bevacizumab are useful in impeding TNBC tumor growth (Figure 2) [54]. The combination of bevacizumab with standard capecitabine or anthracycline/taxane led to an increase in median progression-free survival (mPFS) and OS among patients with locally recurrent or metastatic TNBC. Apatinib plus NACT amazingly increased the PCR rate compared to chemotherapy alone (72.7% versus 50%) [55]. Another study demonstrated that the combination of low-dose apatinib (250 mg/day to 500 mg/day) with chemotherapy is better than chemotherapy alone for metastatic TNBC. Further, the study showed the occurrence of grade 3/4 non-hematologic toxicities was lowered compared to the recommended dose of apatinib (750 mg/day) [56].

PARP inhibitors act efficiently in metastatic TNBC therapy. PARPis trap the PARP1 proteins on the DNA repair intermediates, causing BRCA1/2-defective cells to gather more toxic double-strand breaks (DSBs), leading to cell death [57,58]. For instance, adding the PARPi veliparib to paclitaxel and carboplatin augmented PFS compared to chemotherapy with no change in the toxicity profile in TNBC patients. PFS is the length of time during and after the treatment of a disease such as cancer that a patient lives with the disease but it does not become worse. The PARPi Talazoparib has exhibited improved activity in TNBC when utilized as neoadjuvant monotherapy [59]. In metastatic TNBC therapy, both ipataseritib and capivaseritib have led to encouraging results, mainly in PIK3CA/AKT1/PTEN-reformed tumors (Figure 2) [60,61].

Key proteins signaling DNA damage to cell cycle checkpoints and DNA repair pathways include ATM, ATM, and Rad-3 (ATR) kinases. The stability of the DNA replication fork is linked to the cell cycle; cell cycle checkpoints facilitate the DNA damage signals, activation of signaling pathways, and cell cycle arrest. ATR, Chk1/2, and cyclin-dependent kinase (CDK) are essential in this process. The cell cycle checkpoint proteins (ATR, Chk1/2, WEE1) play crucial roles in stabilizing the replication fork. TNBC therapy targets DNA damage response, particularly utilizing agents that inhibit ATM, ATR, Chk1/2, and WEE1 (Figure 2) [60]. ATR and Chk1/2 inhibitors are prospective therapies for TNBC, interfering with the cell-cycle regulation of TNBC cells (Figure 2). Co-administration of ATR, Chk1/2 inhibitors, and chemotherapeutics inhibit tumor growth in TNBC [60]. Combining PARP inhibitor therapy with cell cycle checkpoint inhibitor therapy can mitigate drug resistance in cancer therapy [62,63].

EGFR inhibitors such as panitumumab are being examined for their effectiveness in initial TNBC. A PCR rate of 42% was recognized by adding panitumumab to NACT with nab-paclitaxel and carboplatin in TNBC patients [64]. Further, the addition of apatinib to NACT greatly improved the PCR rate to 72.7% compared to NACT (50%) (Figure 2) in patients with TNBC [54].

Targeting the PI3K/AKT/mammalian target of the rapamycin (mTOR) pathway in TNBC may be an effective approach. Alpelisib is approved for hormone receptor-positive, HER-2-negative, metastatic breast cancer patients with activating PIK3CA [65]. There have been several clinical trials for targeting the PI3K/AKT/mTOR in TNBC patients. A phase II clinical trial (NCT04216472) for patients with TNBC has been investigating a combination of nab-paclitaxel with alpelisib for the neoadjuvant therapy of anthracycline-refractory (or suboptimal activity) TNBC with PIK3CA or PTEN alterations [57]. This trial studies how effectively nab-paclitaxel and alpelisib work in the treatment of TNBC patients with PIK3CA or PTEN alterations that were anthracycline refractory. The study demonstrated that nab-paclitaxel works in different ways to inhibit the growth of tumor cells, by stopping the cells from dividing and spreading. Alpelisib inhibited tumor growth by blocking some of the enzymes needed for cell growth. Treatment with Alpelisib and nab-paclitaxel reduced the tumor before surgery [66].

Investigations are developing to examine the efficacy of antibody-drug conjugates (ADCs) in TNBC therapy in both pre-clinical and clinical systems [67,68,69]. An ADC contains a mAb and a highly cytotoxic small molecule drug linked via a chemical linker [58,60]. An ADC, Sacituzumab govitecan comprises an anti-Trop-2 antibody joined to the active metabolite of irinotecan, SN-38, through a hydrolyzable linker. The linker allows for SN-38 release intracellularly into TME [67]. Sacituzumab govitecan significantly augmented PFS and median overall survival in relapsed/refractory metastatic TNBC patients [68]. The US FDA has approved progressive ADC drugs such as trastuzumab emtanstine [69]. Sacituzumab govitecan and trastuzumab deruxtecan are in the late phases of clinical trials for metastatic BC including TNBC. Recently, sacituzumab govitecan-hziy (Trodelvy) for adult TNBC patients has been approved by the FDA. Several ADC drugs such as ladiratuzumab vedotin and trastuzumab duocarmizine are in the late phases of clinical trials [69].

Some international guidelines and recommendations are available for the treatment of TNBC depending on the indications [37]. The guidelines suggest the application of chemotherapy for the patient with TNBC following either the NACT or ACT approaches. The use of nab-paclitaxel is still uncertain due to inconsistent outcomes [70,71,72,73]. The AGO guidelines recommend nab-paclitaxel instead of paclitaxel in NACT [74]. The NCCN (National Comprehensive Cancer Network) guidelines suggest soluble taxanes as a replacement for nab-paclitaxel due to hypersensitivity [75]. Anthracycline/taxane-based chemotherapy is the favored choice for early TNBC. Carboplatin was suggested for TNBC regardless of *BRCA* grade. The role of PARP and AKT inhibitors revealed encouraging outcomes in primary TNBC. Adding NACT for high-risk patients is anticipated. After NACT, capecitabine as adjuvant therapy was a good choice in patients with enduring disease [37]. However, chemotherapy is still one of the significant causes of therapy failure in BC and TNBC [76,77].

The therapeutic option favored for early-stage patients is still taxane/anthracycline-based chemotherapy, although they are allied with severe chemoresistance. The 5-year survival rate with localized TNBC is 91%, 65% in cases of locally progressive TNBC, and only 11% with metastatic disease [3,19]. As a result, the selection of TNBC therapy must be personalized and followed by a multidisciplinary targeting attempt. The TNBC therapy is dependent on several factors including tumor size, morphology, histological degree, and *BRCA1/2* grade of the patient [3,78].

The purpose of TILs in TNBC therapy has been examined in pre-clinical and clinical research [79]. The number of TILs in the TME is not sufficient to be effective for TNBC therapy. However, TILs can be extended to a huge number using in vitro cell culture and re-introduced into patients to utilize in TNBC therapy. For example, TIL adoptive cell therapy was developed for the treatment of cancer with high TMB (tumor mutational burden) [80]. Then, neoantigen-specific TILs are enhanced and extended ex vivo and delivered back to the patient. Adoptive cell therapy with TILs has demonstrated promising results in cancer therapy.

IDO1 inhibitors have reached clinical trials for targeting various solid tumors including breast cancers. IDO1is in combination with anti-PD1 antibodies have demonstrated better cooperativity, which could overcome drug resistance and improve the survival rate of patients. However, clinical trials of IDO1i therapy have not yet been successful. As a result, insights into the IDO1 inhibition mechanism and comprehensible clinical trial design are required for IDO1-targeted small molecule drug development for solid tumors including breast cancer [81].

In women, breast cancer is the fourth leading cause of cancer death in the USA. Recently, a major step forward in breast cancer research has been the identification of subtypes with different genetic drivers. Thus, targeted delivery for BC or TNBC therapy has become a crucial step to improving the survival rate.

The cancer survival rate for several cancers, including breast cancer has improved recently [82]. Accordingly, the cancer death rate in the USA dropped 27% between 2000 and 2021. This progress is due to the advances in genome sequencing and screening technologies to provide targeted personalized therapy such as immunotherapy and RNAi therapy. There has been a shift, restructuring the goals in cancer therapy from curing the disease to improving the quality of the patient’s life, turning cancer into a chronic disease rather than a deadly one.

## 4. siRNA-Based Therapies Facilitating an Antitumor Effect in Cancer Therapy

To overcome treatment difficulties, effective targeting of proliferation, cell cycle, angiogenesis and tumor microenvironment, invasion and metastasis, chemoresistance, and enhancing therapeutic performance utilizing siRNA-conjugated nanomaterials and nanobiomaterials in cancer therapy is required. In siRNA-based RNA interference, the oligonucleotides are loaded into nanocarriers before internalization by target cells. When internalized into the target cells, siRNA undergoes endosomal escape before entering the RNAi machinery. After the endosomal escape, the oligonucleotides are merged into the RNA-induced silencing complex (RISC), where they are paired with a target mRNA strand for translational repression or specific gene silencing (Figure 3) [83,84].

Although chemotherapy, radiation therapy, and surgical operation are common approaches in TNBC, these are allied with severe side effects and chemoresistance. Emerging chemoresistance is allied with therapy unsuccess and poor prognosis in TNBC [85]. The application of siRNA therapy has the potential to overcome chemoresistance in TNBC therapy [85]. MDR may evolve in breast cancer cells through different mechanisms [86,87,88,89]. Among them, ABC transporter, modifications of anti-apoptotic genes, and hypoxia are key factors in chemoresistance. ABCB1 (P-glycoprotein) and ABCG2 (BCRP: breast cancer resistance protein) efflux pump members are overexpressed in TNBC tumors [85]. siRNA drug delivery to TNBC tumors using a suitable nanomaterial- or nanobiomaterial-based carrier may be a potential strategy to silence targeted genes to inhibit TNBC tumors and overcome drug resistance in TNBC therapy.

## 5. Targeted siRNA Therapy in TNBC

Targeted siRNA therapeutic is an alternative approach for suitable TNBC therapy. siRNA drugs are capable of silencing diseased genes or proteins that are overexpressed on TNBC cells. However, specific targeting, systemic stability, and delivery efficiency to tumor-diseased cells provide barriers to the utilization of this therapy in the clinical setup. Efficient use of siRNA therapeutic in TNBC requires encapsulation and transport of siRNA with high effectiveness and reduced toxicity. Recent developments in TNBC therapy strategies with siRNA applying nanomaterials and nanobiomaterials as carriers have shown promise in preclinical and clinical settings [85].

MDR in TNBC chemotherapy often occurs due to the overexpression of ATP-binding cassette B1 (ABCB1) protein, which effuses various chemotherapeutic drugs from cancer cells [90]. Downregulation of ABCB1 and ABCG2 by siRNA-based NPs increased the sensitivity to doxorubicin and paclitaxel. Multiple-layered doxorubicin-based NPs decorated with alternate layers of siRNA and poly-L-lysine shielded by an outer shell of hyaluronic acid were developed to target *MDR1*. The doxorubicin–siRNA nanoparticle system inhibited tumor growth in TNBC [90,91].

To reduce chemoresistance in TNBC tumors, targeting anti-apoptotic genes/proteins is crucial. Survivin is an inhibitor of apoptosis and regulates drug resistance [92]. Polymeric micelles of PEG_2000_-PE with an encapsulated anti-survivin siRNA can sensitize TNBC chemotherapy. Administration of anti-survivin siRNA polymer micelles with paclitaxel to MDA-MB-231 cells reduced survivin expression in resistant cancer cells, caused microtubule destabilization, and significantly inhibited tumor growth [93].

ABCB1 siRNA in combination with either paclitaxel or etoposide was tested against TNBC MDA-MB-231 cells. Silencing of ABCB1 by ABCB1 siRNA enhanced the survival of TNBC cells to sub-toxic doses of paclitaxel and etoposide [83]. The study demonstrated that combination therapy of ABCB1 siRNA with paclitaxel and etoposide reduced cell viability to 48.2% and 52.87% of TNBC cells respectively. Colony formation assay indicated that both combinations can inhibit colony formation in TNBC cell lines MDA-MB-231. The combination therapies also reduced TNBC cell migration significantly [94]. Thus, TNBC therapy using ABCB1 siRNA in combination with either paclitaxel or etoposide demonstrated dual effects on inhibition of TNBC cell growth and progression with less cytotoxicity of healthy cells.

An anti-survivin siRNA nanocarrier DLP/siRNA was fabricated with Doxorubicin (D), lycopene (L), and PAMAM (P), and characterized by several techniques. DLP–siRNA inhibited the cancer cell stemness and suppressed tumor development without cardiac toxicity [95] TNBC MDA-MB-231 cells treated with PAMAM–siRNA demonstrated negligible apoptosis at the late apoptosis stage. The MDA-MB-231 cells post-treated with DLP only showed early and late apoptosis at 55.3% and 2.87%, respectively. When MDA-MB-231 cells were treated with DLP-siRNA, 56.3% early apoptosis, and 15% late apoptosis were observed [95]. The effective apoptotic cell death of TNBC cell lines was due to the combined effect of chemotherapeutics and siRNA therapy. The DLP–siRNA system demonstrated efficient cellular uptake and a significant reduction of tumor size in mouse models of TNBC MDA-MB-231 cells (Figure 4).

Forkhead box protein M1 (FOXM1) in humans is encoded by the gene *FOXM1*, which plays a significant role in several molecular processes in TNBC development. *FOXM1* is an effective therapeutic target in TNBC [96]. Administration of liposomal FOXM1 siRNA into TNBC tumors downregulated FOXM1 expression and decreased growth of the TNBC cell line MDA-MB-231 in mice [96,97]. TNBC patients demonstrated an overexpression of eEF2K. Since the FOXM1 and eEF2K are associated with an identical role in TNBC, FOXM1 regulates the expression of eEF2K. Fabricated liposomal siRNA NPs upon binding to the promoter region of eEF2K controlled its expression. Altered gold anti-eEF2K siRNA NPs were very effective as a therapeutic of TNBC tumors in mice xenografts, ensuing knockdown of eEF2K expression and reduction of tumor development and spreading [96,98]. Further, eEF2K inhibition improved mitochondrial function and reduced oxidative stress [99]. Therefore, eEF2K inhibition by targeted siRNA therapy is an innovative approach for TNBC therapy.

*MDM2* gene deactivates tumor suppressor p53, which is highly expressed in TNBC tumor cells. *MDM2* overexpression leads to repression of p53 causing proliferation of tumor cells and reduction in apoptosis. Single-walled carbon nanotubes (SWCNTs) delivered siRNA effectively to target *MDM2*. PEG-modified SWCNTs were effective in *MDM2* silencing, reduced proliferation, and enhanced apoptotic cell death in breast tumors [100].

Combination therapy of p53 siRNA and epigallocatechin-3-gallate (EGCG) reduced the stimulation of anti-apoptotic genes related to therapy resistance, such as BAG3, XIAP, and RIPK2, which may enhance the sensitivity of cancer cells to therapy in TNBC tumors. Integration of p53 siRNA and EGCG improved the antitumoral effect on the TNBC cancer cells by activation of apoptosis and autophagy. This study demonstrated a potential TNBC therapy strategy targeting a genetic component such as overexpressed p53 and delivering p53siRNA in conjunction with a natural anticancer drug EGCG [101].

NF-kB is a key controller of TNBC. In normal cells, NF-kB is bound to an inhibitor of kB, IkB. In cancer cells, inhibitory kB kinases (IKKs) are stimulated to phosphorylate IkBs and tempt their decomposition leading to the translocation of NF-kB to the nucleus. Activated NF-kB enhances cell proliferation, invasion, and metastasis, and reduces apoptosis in breast cancer. IKKℇ is one of the five family members of IKKs also known as IKBKE. Targeting IKBKE with siRNA is a prospective therapeutic approach for TNBC patients. A hybrid nanosystem to co-deliver the IKBKE siRNA and cabazitaxel was fabricated to inhibit the TNBC tumors. The nanocomplex was modified with HA to target CD44 on TNBC cells [102]. Encapsulation of cabazitaxel in peptide-based hybrid siRNA nanoplatforms enhances the function of siRNA in TNBC tumors [102]. As a result, a peptide-based nanosystem incorporating siRNA and chemotherapeutics is a promising diverse platform with enhanced cellular uptake, and improved tumor penetration for effective TNBC therapy.

CDK11 and CK2 protein kinases are overexpressed in TNBC tumor cells. Nanocapsules coated with Tenfibgen and containing anti-CDK11siRNA/anti-CK2 siRNA were investigated for targeting Tenascein-C-receptors found in breast cancer stroma. Anti-CDK11siRNA/anti-CK2 siRNA demonstrated efficiency in targeting tumor cells, inhibiting tumor development and proliferation [103]. *c-Myc* and *CDK1* play significant roles in promoting cell death and apoptosis. Downregulation of *c-Myc* and *CDK1* genes resulted in specific targeting and death in TNBC tumor cells. *c-Myc* gene knockdown by anti-*c-Myc* siRNA-encapsulated NPs caused *c-Myc* silencing with minor effects on TNBC cells, while siRNA drug delivery targeting CDK1 resulted in enhanced inhibition of TNBC cells with more *c-Myc* expression. Thus, combination therapy with *c-Myc-* and CDK1-targeting siRNA drugs is effective for reduced cell proliferation, apoptosis, and inhibition of TNBC tumors [103,104].

*Myc* and *MCL1* are often co-amplified in drug-resistant TNBC cells after neoadjuvant chemotherapy [105]. Further, *Myc* and *MCL1* increased mitochondrial oxidative phosphorylation (mtOXPHOS) and generation of ROS, which regulate drug-resistant CSCs. A bi-product of stimulated mtOXPHOS is increased levels of ROS, which lead to the accumulation of HIF-1α. Further, HIF-1α is related to CSCs enrichment in TNBC. Myc and MLC1 confer chemotherapy resistance by expansion of CSCs via mtOXPHOS in TNBC tumors. Targeting mitochondrial respiration and HIF-1α may inhibit chemotherapy resistance in TNBC. Therefore, inhibition of *Myc* and *MCL1*-targeting siRNA may be a potential approach to inhibit drug-resistant TNBC tumors [105].

Recently, the antiapoptotic gene *MCL1* was targeted using a lipid–siRNA conjugate suitable for systemic serum albumin binding [106]. Divalent lipid-conjugated modified siRNAs were utilized for serum-albumin-binding and enhanced TNBC tumor delivery. Systemic modification of the siRNA–conjugate structure facilitated intracellular siRNA delivery through binding to serum albumin. The lead structure enhanced tumor siRNA accumulation 12-fold in orthotopic TNBC tumors over the parent siRNA. The study demonstrated ~80% silencing of the *MCL1* oncogene with improved survival resulted in TNBC models compared to an *MCL1* small molecule inhibitor. Therefore, optimization of the structure of siRNA conjugates for systemic albumin binding demonstrated effective *MCL1* targeting and improved TNBC therapy [106].

Polo-like kinase 1 (PLK1) upregulation is observed in primary and metastatic TNBC tumors. Downregulation of *PLK1* enhances tumor cell death in metastatic TNBC tumors [107,108]. A targeted nanoplatforms with anti-*PLK1*siRNA wrapped within an antibody-conjugated bioreducible cross-linked with PEI and PEG layer by layer with coated mesoporous silica nanoparticles (MSNP) was efficient in metastatic TNBC therapy. It was observed that MSNP functioned as ROS inhibitor, and reduced metastasis. Further, anti-*PLK1*siRNA exhibited tumor-cell-killing effectiveness through apoptosis [109].

A lipid NP encapsulating siRNA, decorated with a Fab′ antibody against heparin-binding EGF-like growth factor, αHB-EGF LNP siPLK1 was developed for targeting *PLK1* in MDA-MB-231 cell lines overexpressing HB-EGF [110]. αHB-EGF LNP siPLK1 effectively delivered siPLK1 to tumor cells in MDA-MB-231 cancer-bearing mice. The PLK1 protein in the mice was effectively suppressed after i.v. injection of αHB-EGF LNP siPLK1. αHB-EGF LNP siPLK1 significantly inhibited tumor growth in the mice indicating that αHB-EGF LNP siPLK1 is a promising therapy for TNBC cell lines expressing HB-EGF [110].

Twist-related protein (*TWIST*) stimulates EMT, promotes cancer stem cells, and reduces apoptosis, hence, increasing the risk of disease relapse with poor prognosis in TNBC patients. PAMAM dendrimer loaded with TWIST-siRNA was constructed for siRNA delivery to TNBC cells, which decreased TWIST expression along with phenotypic variations, and reduced cancer cell movement, diminishing TWIST-inspired amplification of mesenchymal marker, inhibiting TNBC tumor cells [111,112].

TP53 gene mutation is frequent in TNBC, which is associated with hemizygous loss of *POLR2A* [103]. TNBC cell lines allied with hemizygous deletion of *POLR2A* are susceptible to nanotherapeutics. Silencing of POLR2A using a pH-sensitive nanobomb incorporated with a *POLR2*AsiRNA (siPol2) and guanidine–CO_2_ functionalized chitosan at neutral pH led to an effective growth inhibition of TNBC tumor characterized by hemizygous loss of *POLR2A*. A siRNA nanoplatform containing CG-CO_2_, siPol2@NPs was effectively internalized into TNBC cells and caused selective cell death. Herein, the siRNA-nanoplatform-induced silencing of *POLR2A* killed TNBC cells efficiently [99]. The study identified *POLR2A* in the *TP53*-neighbouring region as a collateral vulnerable target in TNBC tumors suggesting the siRNA nanoplatform containing CG-CO_2_, siPol2@NPs, as an amenable strategy for TNBC targeted therapy [103].

RhoA and RhoC are low molecular weight GTPases of the RAS family. RhoA and RhoC are overexpressed in TNBC tumors [113]. Silencing of RhoA with an anti-RhoA siRNA using a chitosan-coated polyisohexylcyanoacrylate (PIHCA) nanocarrier system silenced RhoA, effectively causing TNBC cell death without toxicity, demonstrating an efficient therapeutic approach [113].

Lipocalin 2 (Lcn2) protein is upregulated in TNBC cells. A molecular target for TNBC, intracellular adhesion molecule (ICM1)-targeted siRNA delivery demonstrated effective Lcn2 knockdown. An ICM1–Lnc2-siRNA nanoplatform was used to internalize siRNA into the TNBC cell line MDA-MB-231, which effectively silenced Lcn2, reducing angiogenesis and metastasis in TNBC tumors [101]. An effective Lcn2 knockdown by Lnc2-siRNA-encapsulating liposomal formulation, ICM1–Lnc2-siRNA led to a noteworthy inhibition of VEGF from MDA-MB-231 cells, resulting in reduced angiogenesis both in vitro and in vivo [114].

A recent study demonstrated that the polyplexes fabricated from 30-cholesterol-modified siRNA, Chol-siRNA, or Chol-DsiRNA and cationic PLL [30]-PEG_5K_ enhanced the effectiveness of RNAi in 4T1 metastatic TNBC tumors against STAT3. Chol-siSTAT3 and Chol-DsiSTAT3 also suppressed mRNA in 4T1 cells in a similar manner. As a result, Chol-siRNA and Chol-DsiRNA are prospective aspirants for RNAi therapy of breast cancer in the clinic [115].

siRNA-loaded exosomes are a promising nanodelivery platform for breast cancer therapy. Natural exosomes, engineered exosomes with altered membranes and enhanced cargo loading capacity, and stable bionic exosomes are excellent nanoplatforms for drug delivery and are considered next-generation nano theragnostic for BC and TNBC. Conjugation of exosome membrane to siRNA produces a suitable construct with an effective loading ability [116,117]. For example, S100A4 is a metastasis-promoting protein. Silencing S100A4 using an exosome-loaded S100A4-siRNA nanoplatform reduced TNBC cell growth and metastatic progression effectively [118].

X-box-binding protein 1 (XBP-1) is involved in TNBC chemoresistance and disease development. Downregulation of XBP1 using XBP1siRNA decreased angiogenesis, hindered cell proliferation, and inhibited breast cancer growth by improving sensitization of chemotherapy [119,120]. Systemic delivery of aptamer-conjugated XBP1siRNA, 3WJ-HER2apt-siXBP1 NPs exhibited efficient XBP1 silencing, resulting in breast cancer suppression. XBP1 recognition by the NPs impaired angiogenesis and inhibited cell proliferation, suppressing breast cancer growth effectively, and sensitizing chemotherapy in a breast cancer mouse model [120]. In another study, RNase-resistant siRNA NPs were developed from the Phi29 DNA packaging motor to improve targeted TNBC therapy and sensitize TNBC cells to chemotherapy. The siRNA NPs were fabricated with EGFR targeting aptamer and XBP1-siRNA. This aptamer-conjugated XBP1siRNA effectively deleted XBP1 expression and suppressed TNBC tumor growth after intravenous administration, promoted TNBC sensitization to chemotherapy, and impeded angiogenesis in vivo [121]. This could be an effective strategy for TNBC therapy [119,120].

TG-interacting factor (TGIF) knockdown is a therapeutic target for preventing and targeting chemoresistance and metastasis in TNBC patients. TGIF knockdown inhibits Wnt target genes and tumor invasion. TGIF protein expression silencing using TGIF siRNA sensify the TNBC cell line MDA-MB-231 to cisplatin-induced apoptosis [122]. Reducing TGIF protein using siTGIF prompted apoptotic stimulation of cisplatin to MDA-MB-231 cells, demonstrating that TGIF-targeted therapy via siRNA delivery using a suitable nanocarrier may be used for sensitizing a chemotherapy response in TNBC therapy [122].

A recent study formulated a nanoshell (NS) combination using silica core–gold shell NPs, Frizzled7 (FZD7) antibody, orthopyridyl disulfide-PEG-valerate (OPSS–PEG–SVA), and β-catenin siRNA for TNBC therapy. The NS combination, Si–Au–OPSS–PEG–FZD7-siRNA, was delivered to TNBC MDA-MB-231 cells [123]. The developed NS combination enabled receptor and effector level targeting of TNBC cells through specific binding of FZD7 antibody to TNBC cells and inhibiting Wnt signaling by locking FZD7 receptors on TNBC cells, while siRNA downregulated β-catenin. NS coated with FZD7 antibody and β-catenin siRNA downregulated Wnt-related genes (C-Myc, CCND1) more efficiently than either antibody-coated or siRNA-coated NS alone, leading to enhanced inhibition of cell proliferation, spheroid building, and migration. These NS combinations with dual FZD7 antibody/siRNA nanocarrier effectively inhibited tumor development, brain metastasis, and disease recurrence. The study exhibited significant downregulation of Wnt signaling at both receptor and mRNA levels through the antibody/siRNA NS combination, which could be a promising approach for TNBC therapy [123].

The transient receptor potential cation channel, subfamily C, and member 6 are known as a TRPC6 gene encoding a protein also known as TRPC6. TRPC6 plays a dynamic role in the activation of Ca^2+^ entry into breast cancer cells with no effect in normal breast cells, such as MCF-10. siTRPC6 subdued MCF-7 and TNBC cell line MDA-MB-231, and reduced cell migration and proliferation. Thus, TRPC6 can be effectively silenced by the siTRPC6 system. These outcomes exhibited an innovative approach for controlling the Ca^2+^ influx into cancer cells and the evolution of biomarkers for TNBC cells [124]. A siTRPC6 delivery system, Shamporter (sheep antimouse podocyte transporter) was developed for silencing TRPC6 levels in an animal model. Shamporter coupled with siTRPC6 was injected via the tail vein into normal rats and substantially silenced the TRPC6 protein level [125]. This approach may be implicated in TNBC therapy for efficient silencing of TRPC6.

A key protein involved in cancer is EGFR. An EGFR-targeted siEGFR was transfected into the TNBC cells via cell-penetrating peptide (CPP)–nanobubbles synergized with an ultrasound-mediated microbubble destruction strategy [126]. CPP and siEGFR were loaded into the shells of nanobubbles. In vivo, experiments indicated the downregulation of EGFR mRNA and protein significantly in a xenograft tumor derived from TNBC cells. CPP–NBs–siEGFR was found to have potential and promise for TNBC therapy [126].

Targeted nanocarriers with specific humanized antibodies or aptamers are an important class of nanobiomaterials for siRNA-based therapeutics [127,128,129,130]. A targeted nanobiomaterial with humanized anti-EGFR scFV, NM-scFV, was developed as a siRNA delivery system for TNBC cells using cationic polymers such as chitosan and poly-L-arginine as a siRNA complexing agent. NM–scFV was based on superparamagnetic nanoparticles (SPION). This biomaterial–siRNA-based delivery system was effective in delivering siRNA to TNBC cell lines that exhibited significant gene silencing and antitumor effects [126].

Recent studies utilized aptamers to distinguish and target TNBC cells [128,129]. QD-lipid nanocarriers (QLs) were used to fabricate anti-EGFR aptamer–lipid conjugates (aptamo-QLs) for TNBC targeting. In this study, TNBC-targeting aptamo-QLs and anti-EGFR antibody-coupled immuno-QLs were used as siRNA delivery platforms in both in vitro and in vivo settings. Both types of EGFR-targeted QLs increased delivery to target tumor cells with more efficient gene silencing and tumor imaging than non-targeted control QLs. Further, combination therapy with Bcl-2 and PKC-I siRNA loaded into the anti-EGFR QLs significantly hindered tumor growth and metastatic spread [128].

Oncogenic miRNA, miR21 overexpressed in TNBC demonstrating a potential target for TNBC therapy [131]. miRNA can interplay with genes. miR21 can interact with AKT protein, promoting TNBC growth. A recent study utilizing G11-peptide-conjugated siRNA (5′-UCAACAUCAGUC UGAUAAGCUA-3′)–chitosan NPs (GE11–siRNA–CSNPs) was assessed for targeting EGFR overexpression in TNBC. The study demonstrated significant selective inhibition of miR21 in TNBC tumors [24]. GE11–siRNA–CSNPs significantly inhibit miRNA-21 expression, cell migration, and colony formation. The results also indicated that GE11–siRNA–CSNPs impeded cell cycle progression. It induces cell death by reducing the expression of the antiapoptotic gene Bcl-2 and increasing the expression of the proapoptotic genes Bax, Caspase 3, and Caspase 9. Additionally, docking analysis and immunoblot investigations verified that GE1–siRNA–CSNPs, which specifically target TNBC cells and suppress miRNA-21, can prevent the effects of miRNA-21 on the proliferation of TNBC cells via controlling EGFR and subsequently inhibiting the PI3K/AKT and ERK1/2 signaling axis. EGFR downstream Ras/Raf/MEK/ERK and PI3K/AKT signaling pathways are significant cell proliferation pathways in TNBC. As a result, inhibition of the PI3K/AKT and ERK1/2 signaling axis demonstrated significant inhibition of the TNBC MD-MB-231 cell line [132]. The GE11–siRNA–CSNPs designed, which specifically targeted TNBC cells, offer a novel approach to treating breast cancer with improved effectiveness. This study suggested that GE11–siRNA–CSNPs could be a promising candidate for further assessment as an additional strategy in treating TNBC [24]. Further, combination therapy using GE11–siRNA–CSNPs and ERK inhibitor U0126 significantly inhibited ERK1/2 demonstrating more suppression of TNBC MD-MB-231 cells [24].

An aptamer-induced siRNA NP directing CD44 expression in TNBC tumors was formulated for TNBC therapy [133]. The NPs’ core consisted of a siRNA–protamine complex, and the shell comprised an aptamer ligand for targeting the expression of CD44 on TNBC tumor cells. The study exhibited that the drug targeting TNBC tumors significantly enhanced anticancer effects [133,134].

TNBC patients demonstrate mutations in the PI3K/mTOR pathway associated with chemoresistance [135]. The protein kinase mTOR works with two discrete protein complexes, mTORC1 and mTORC2. mTORC1 resistance in TNBC is not an effective approach for therapy; however, mTORC2 regulates cell survival, metastasis, and motility in several cancers [135]. A strategy was developed to inhibit the mTORC2 obligate factor, Rictor, in TNBC cell line MDA-MB-231. A polymeric and PEG-based si-NP encapsulating siRNA against Rictor, 50B8–DP100 siNP (siRictor NPs) was fabricated and investigated in orthotopic MDA-MB-231 tumor-bearing mice [134]. Developed siRictor NPs caused 80% tumor Rictor silencing in orthotopic MDA-MB-231 tumor-bearing mice. siRictor NPs significantly inhibited the mTORC2 activity in TNBC cell lines. Treatment with siRictor NPs in an orthotopic HCC70 TNBC model reduced cell growth through enhanced apoptosis. siRictor NPs reduced cell growth by 52% compared to siControl NPs in TNBC cell lines [135]. This work demonstrated the potential of Rictor knockdown therapy in mTOR/PI3K-active TNBC tumors.

Lipopeptide-encapsulated Notch1-silencing siRNA nanocomplexes (Lipopeptide-siRNA complexes) were developed and tested against HUVEC and the TNBC MDA-MB-231 cell line [120]. Fabricated lipopeptide has an Arginine–Sarcosine–Arginine segment for providing protease stability, minimizing adjacent arginine–arginine repulsion, and reducing intermolecular aggregations and a-tocopherol as the lipidic moiety for enabling cellular permeability. Lipopeptide–siRNA complexes significantly inhibited HUVEC and MDA-MB-231-cell-mediated pre-metastatic niche formation in metastasis [136]. The study showed that a combination therapy with mTOR inhibitor metformin and Notch1-silencing-lipopeptide–siRNA complexes has demonstrated a synergistic effect. Engineered lipopeptide-based siRNA therapeutics inhibited metastasis and TNBC stemness in vitro and in vivo in a zebrafish model [136].

Immune checkpoint blockade (ICB) by anti-programmed cell death 1 (aPD-1) antibody has been implicated in TNBC therapy. However, the enhancement of ICB responsiveness is a crucial factor in TNBC therapy to make the therapy effective [137]. Ataxia telangiectasia mutation (ATM) is a crucial factor in the DNA damage response (DDR) pathway, which is allied with tumor growth. ATM can control the tumor immune microenvironment, affecting the ICB response. Reducing ATM can increase ICB responsiveness by promoting mitochondrial DNA cytoplasmic escape and stimulating the innate immune signaling pathway [137]. A nanoparticle, 1,2-dioleoyl-glycero-3-phosphatidylcholine (DOPC) liposome was designed and fabricated to deliver ATM siRNA (siATM) to TNBC cells to test the consequence of siATM on the ICB sensitivity of TNBC. Both in vitro and in vivo studies found that DOPC–siATM enhanced the siRNA delivery to TNBC tumors and effectively reduced the ATM proteins [137]. The study demonstrated that siATM could stimulate cytotoxic T lymphocytes and control the immune-responsive tumor microenvironment through activation of the cGAS-STING pathway [137]. Therefore, combining siATM and ICB may be a promising therapy approach for TNBC.

DANCR is an oncogenic long noncoding RNA (lncRNA) overexpressed in TNBC. Previously, targeted multifunctional ionizable lipid ECO/siRNA NPs were effective in the downregulation of lncRNA in TNBC [138]. A dual-targeted ECO/siDANCR NP was developed for targeting a tumor extracellular matrix oncoprotein, extradomain B fibronectin (EDB-FN), and integrins overexpressed on TNBC cells by boosted delivery of siDANCR [139]. The study demonstrated that treatment of Hs578T TNBC and ER-positive MCF-7 cells in vitro caused noteworthy downregulation of DANCR and EDB-FN and reduced invasion and 3D spheroid formation of cells. The study utilized magnetic resonance molecular imaging (MRMI) with MT218, an EDB-FN-targeted contrast agent to study targeted ECO/siDANCR NPs therapy in female nude mice bearing orthotopic Hs578T and MCF-7 xenografts, respectively [139]. MRMI indicated that dual-targeted ECO/siDANCR NPs resulted in noteworthy tumor growth inhibition in both models and decreased EDB-FN expression in TNBC cell lines significantly. Therefore, dual-targeted ECO/siDANCR therapy has the potential for TNBC as well as MCF-7 cell lines [139].

siRNA therapies in TNBC have the advantage of silencing multiple genes, inhibiting tumors, and reducing drug resistance [23]. Several siRNA candidates (siRNA1-7) were tested against the TNBC MDA-MB-231 cell line. In the study, siRNA 7 was identified as a potential candidate to reduce TNBC tumors in mouse xenografted model silencing multiple genes, PRKCE, RBPJ, ZNF8737, and CDC7 in the TNBC MDA-MB-231 cell line [23]. The target genes in MDA-MB-231 were identified with the genome-wide search of seed sequences. siRNA 7 demonstrated a significant reduction in cell viability in BT 474 and MDA-MB-231 compared to MCF-7 cell lines (Figure 5). This study indicated the silencing of multiple genes by targeted delivery of siRNA specifically in TNBC cells [23].

Several molecular factors including MDM2, CDK11, CK2, TWIST, c-Myc, PLK1, MCL1, survivin, mTORC2, EGFR, CXCR4, and Lnc2 are overexpressed in TNBC cells (Figure 6) [105,106,139,140,141,142,143,144,145,146,147,148,149,150,151]. Silencing these molecular factors using siRNA-induced bionanoplatforms enhanced the inhibition of tumor development. Many preclinical studies exhibited that hindrance of any of these factors applying siRNA-based gene therapy is potential for inhibition of cell proliferation and survival of TNBC patients [23,24,95,123]. To date, numerous pre-clinical investigations have been applying siRNA therapeutics in nanoformulations for enhanced therapy of TNBC patients [85,105,106,130,151,152,153,154,155]. There have been many efforts to promote an anticancer effect mediated by the use of a siRNA-drug-based TNBC therapy in preclinical settings (Table 3). Recently, several siRNA drugs have received approval from the US FDA for use in the treatment of cancer and other diseases, which influenced researchers to find a solution for deadly TNBC treatment utilizing targeted siRNA therapy that is capable of silencing effectively molecular-level specific targets in TNBC tumorigenesis.

Because of the inadequate targeted therapy for TNBC patients, chemotherapy, radiation therapy, and surgical resections are frequently utilized therapy approaches. As TNBC is sensitive to chemotherapy and easy to apply, it is a common therapy option for TNBC patients. Advancements in targeted therapy of TNBC such as PARP inhibitors, PI3K/AKT/mTOR inhibitors, RNase inhibitors, AR antagonists, antibody-drug conjugates, and immune checkpoint inhibitors have proven to be innovative treatment approaches [156,157]. Alternative siRNA-based strategies are needed due to chemotherapy resistance, and terrible side effects from radiation therapy and surgical operations on tumors. Suppressing significant molecular factors linked to TNBC utilizing siRNA-induced nanobiomaterial systems is effective in controlling tumor growth [158]. Many pre-clinical and clinical studies demonstrated that utilizing siRNA-based nanoparticles is effective in inhibiting cell proliferation and migration and improves survival in TNBC. Therefore, using nanobiomaterial–siRNA-based therapies that silence or downregulate molecular factors involved in TNBC may be effective in TNBC therapy.

Since systemic use of naked siRNA suffers from several obstacles including inefficient systemic delivery, intracellular trafficking, and immune stimulation, biomaterial-induced siRNA-based therapeutics are used for the systemic delivery of siRNA. Nanomaterials and nanobiomaterials are crucial for siRNA delivery platforms to overcome the systemic pitfalls, an approach that offers specific and selective siRNA-based therapeutics, effective internalization into the tumor cells, adequate intracellular retention, and efficient anticancer function with little or negligible side effects. Nanobiomaterial-assisted siRNA delivery platforms effectively encapsulate siRNA, protecting it while targeting the tumor cells and escaping the immune stimulation [85]. Nanobiomaterial–siRNA platforms are safe and well tolerated and are associated with insignificant bodyweight changes and immune cytokine production [159]. However, because of high intra- and inter-tumoral heterogeneity and metastasis in TNBC, nanoparticle-assisted siRNA therapeutics may not be highly effective as a single modality [8]. The incorporation of multiple drug modalities such as chemotherapeutics or a PTT agent with siRNA-based drugs in one nanobiomaterial carrier may offer a significant antitumor effect compared to a single agent or sequential delivery of multiple agents [85].

siRNA therapeutics for TNBC patients may be effective and promising as it can downregulate specific genes overexpressed in TNBC tumors [23,24,95,123]. Many siRNA-induced NPs are in different phases of clinical trials or have been approved by the US FDA for therapy of various types of cancers [85,160]. There are some siRNA therapies in clinical trials such as TKM-PLK1 (TKM-080301) and DCR-MYC targeting *PLK1* and *Myc* genes, respectively, in solid tumors that also overexpress in TNBC [85]. These approaches might be applied to TNBC therapy. Recently, four siRNA drugs have received approval from the US FDA: patisiran in 2018, givosiran in 2019, lumasiran in 2020, and inclisiran in 2021 [130]. As a result, siRNA drugs have lately received enhanced consideration. Fabrication of an appropriate siRNA-induced delivery platform is crucial for cancer therapy including TNBC in clinical setting. These strategies should help impede TNBC and improve the survival of the patients.

## 6. Conclusions and Future Perspective

siRNA therapy has advanced into one of the important frontiers in cancer therapy. Innovative nanomaterial- and nanobiomaterial-assisted siRNA therapeutics delivery has greatly improved BC and TNBC therapy in both pre-clinical and clinical settings. Effective silencing of the crucial molecular factors associated with TNBC diseases using siRNA-mediated nanobioplatforms to prompt suppression of tumor development is vital for TNBC therapy. Controlling these factors using siRNA-based nanomaterial-mediated therapy is necessary for the reduction of cell proliferation, tumor growth lessening, and improved survival in TNBC patients. Downregulation or silencing of molecular factors including MDM2, CDK11, CK2, TWIST, c-Myc, PLK1, MLC1, survivin, mTORC2, EGFR, Wnt, and CXCR4 by siRNA therapies may be effective in TNBC therapy. According to recent cancer statistics, cancer survival rates for several cancers including breast cancer are improving in the USA. The cancer death rate in the USA has dropped 27% over the past 20 years. This progress is due to the improvements in the technologies that advance targeted, personalized therapy such as nanomaterial- and nanobiomaterial-assisted siRNA therapy and immunotherapy. As a result, there has been a restructuring of the goals in cancer therapy from outright curing the disease to a focus on the quality of a patient’s life that turns cancer into a chronic, survivable disease rather than a fatal one using a combination therapy approach employing novel nanoparticle-assisted siRNA-based therapy, chemotherapy, and immunotherapy as highly effective therapies against BC and TNBC. In recent years, the approval of four siRNA drugs patisiran, givosiran, lumasiran, and inclisiran by the US FDA raises new hope for the development of novel nanomaterial- and nanobiomaterial-induced siRNA therapies for highly aggressive otherwise undruggable cancers including TNBC. Molecularly designed novel multifunctional biodegradable nanocarriers with adequate stability to overcome extra- and intra-cellular barriers to systemic siRNA delivery will further improve TNBC therapy. Moreover, the development of smart nanocarriers to deliver two or more different types of siRNA simultaneously may be able to silence multiple genes at a time to enhance the therapy of highly aggressive TNBC and reduce the mortality rate improving the quality of life of TNBC patients.

## Figures and Tables

**Figure 1 bioengineering-11-00830-f001:**
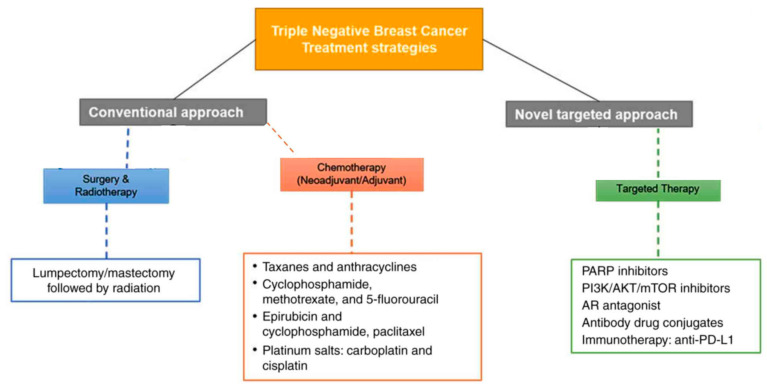
Various methods of treatment in TNBC therapy. (Adapted from [4] in modified form).

**Figure 2 bioengineering-11-00830-f002:**
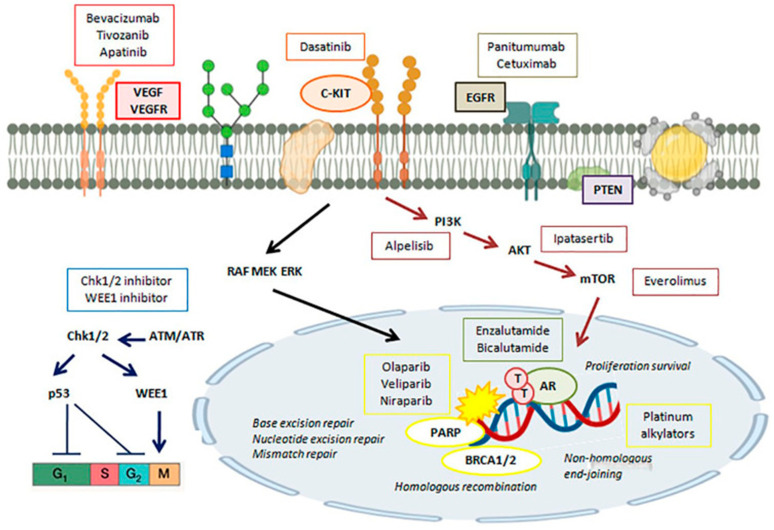
Some therapeutic targets (VEGF, VEGFR, C-KIT, EGFR, PTEN, Chk1/2, WEE1, PARP, BRCA1/2, mTOR) and their inhibitors in TNBC therapy. (Adapted from [3]).

**Figure 3 bioengineering-11-00830-f003:**
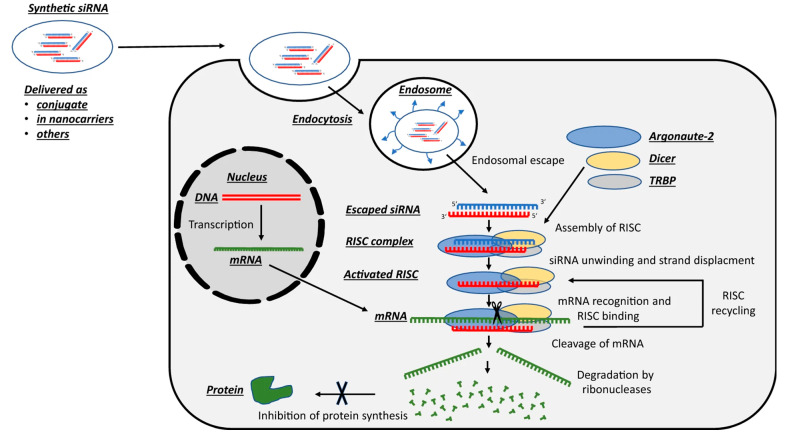
Mechanism of RNA interference in siRNA therapeutics for gene silencing. Adapted from [84].

**Figure 4 bioengineering-11-00830-f004:**
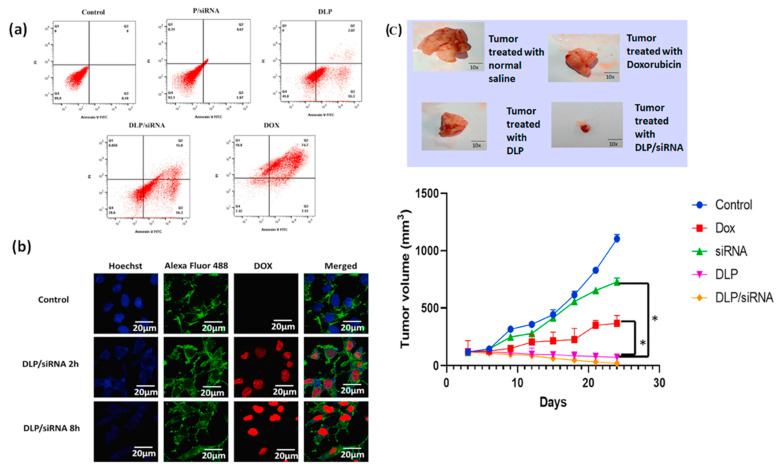
Flow cytometry evaluation of apoptotic cell death with DLP/siRNA indicating cell populations in Q1, Q2, Q3, and Q4 phases (clockwise) (**a**) showed cellular uptake of DLP siRNA (after 2 h and 8 h) (**b**) and inhibition of tumor cells by DLP/siRNA in a mouse model (**c**) (Adapted from [95] with permission).

**Figure 5 bioengineering-11-00830-f005:**
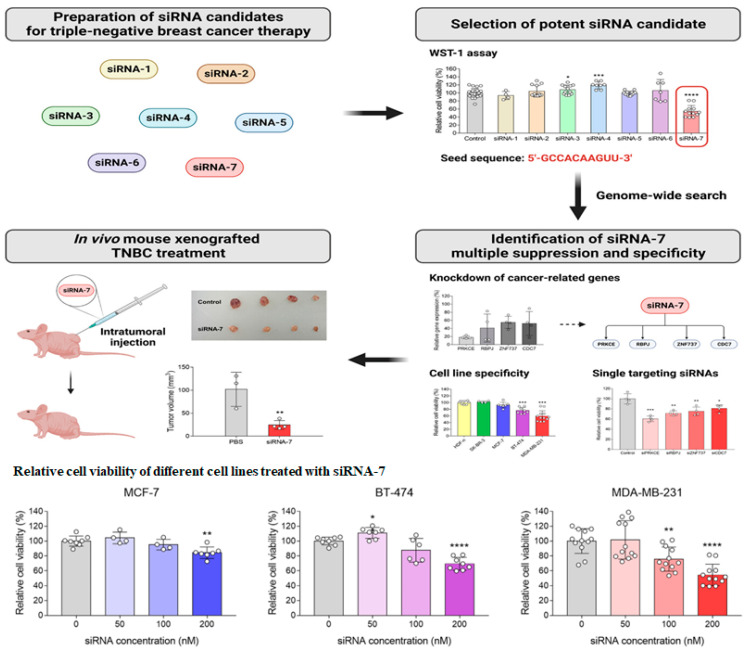
Identification of effective siRNA therapeutics for selective silencing of multiple genes (**top**) and relative cell viability of different breast cancer cell lines at different siRNA concentrations (**bottom**). (Adapted from [23] with permission).

**Figure 6 bioengineering-11-00830-f006:**
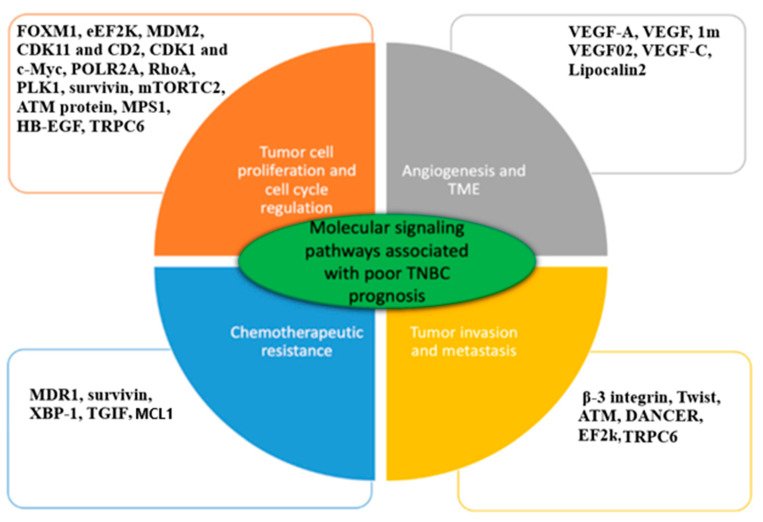
Key genes, proteins, and TFs are intricate in molecular pathways allied with low TNBC prognosis. These factors are upregulated in TNBC tumors. Engineered siRNA-based anticancer drugs targeting single or multiple genes/proteins/TFs caused downregulation and suppression and significantly lowered disease progression. (Adapted from [85] in modified form).

**Table 1 bioengineering-11-00830-t001:** The genes associated with TNBC development and breast cancer risk factors.

Gene	Associated with TNBC	Breast Cancer Risk Factor	Clinical Guideline
Ataxia-telangiectasia mutated (*ATM*)	Likely yes	Two- to three-fold	National Comprehensive Cancer Network (NCCN): screening breast MRI
*BARD1*	Likely yes	-	-
*BRCA1*	Yes; high prevalence of 185delAG and 5382insC founder mutations in TNBC	10-fold	American Cancer Society (ACS) and NCCN: Screening breast MRI, recommending risk-reducing bilateral Salpingo-Oophorectomy (BSO), discussing risk-reducing mastectomy
*BRCA2*	Yes	10-fold	ACS and NCN: Screening breast MRI, recommending (BSO), discussing risk-reducing mastectomy
*MR*	Likely yes	-	-
*NBN*	Likely yes	Two- to three-fold	-
*PALB2*	Yes	Three- to five-fold	NCCN: Screening breast MRI, discussing risk-reducing mastectomy
*PTEN*	Likely yes	Five-fold or more	NCCN: Screening breast MRI, discussing risk-reducing mastectomy
*RAD50*	Yes	-	-
*RAD51C*	Yes	-	NCCN: Considering BSO
*RAD51D*	Yes	_	NCCN: Considering BSO
*XRCC2*	Likely yes	-	-

(Adapted from [35] in modified form).

**Table 2 bioengineering-11-00830-t002:** Some prognostic and predictive biomarkers in TNBC.

Biomarker	Prevalence in TNBC	Mechanism	References
BRCA 1/2 germline mutations	10–20%	Homologous recombination and DNA double-strand break repair	[35,36,37]
Elevated HRD score	40–70%	Homologous recombination and DNA double-strand break repair	[41]
PD-L1	Variability (immune vs. tumor), disease stage, antibody: 40% on immune cells (SP142 antibody) in metastatic disease, 80% by CPS ≥ 1 (22C3) in primary disease	Evasion of tumor immune surveillance	[27,28]
TILs	Variability (intra-tumoral vs. stromal, primary vs. metastatic)	Stromal lymphocytic infiltration of tumor microenvironment	[33,42,43]
High tumor mutational burden	3–11%	Somatic mutations per megabase of DNA	[33]
AR (androgen receptor)	30–35%	Steroid nuclear transcription factor	[45]
EGFR	13–76% EGFR1 overexpression: 18% EGFR1 gene amplification 33% EGFR2 gene amplification: <5%	Receptor tyrosine kinase involved in cell proliferation/survival	[46,47]
VEGF	30–60%	Bind to receptor tyrosine kinase and promote angiogenesis	[45]
*TP53* mutations	80%	Encodes transcription factor protein that promotes cell cycle arrest	[48,49,50]
PI3K/AKT/mTOR	*PI3K* 7–9%, *PTEN* 30–50%	PI3K: intracellular lipid kinases in a signaling cascade that promote cell proliferation/activate survival, PTEN: tumor suppressor gene that downregulates signaling cascade	[45,51]
*NTRK* gene fusion	<1%	Gene fusion results in constitutively active TRK proteins which promotes tumor growth	[45]
Notch signaling	10%	Oncogenes involved in cell proliferation, cell death, cell differentiation, and stem cell maintenance	[52,53]

**Table 3 bioengineering-11-00830-t003:** Anticancer effect mediated by siRNA-drug-based TNBC therapy in preclinical settings.

Target Gene or Protein	siRNA-Conjugate with NPs	TNBC Cell Lines	Anticancer Effect	References
MDM2	PEG-functionalized SWNTs	Breast cancer B-cap 37	*MDM2* silencing, reduced proliferation, and enhanced apoptotic cell death in breast tumors	[100]
FOXM1	Liposomal NPs	MDA-MB-231	Downregulated FOXM1 expression and inhibited cell-cycle regulation, migration/invasion, and survival, decreased growth of the TNBC cell line, MDA-MB-231 in mice. Also downregulated eEF2K expression in TNBC tumors.	[96,98,99]
FOXM1	PEI–cationic polymer	MDA-MB-231	Reduced FOXM1 protein expression level in TNBC tumor	[97]
PLK1	Mesoporous silica NPs	MDA-MB-231, BT549	Suppressed PLK1 proteins in TNBC xenografted mice and reduced tumor growth. Inhibited ROS, induced apoptotic cell death in TNBC, and reduced metastasis.	[107,108,109,110]
RhOA	Chitosan-coated PIHCA NPs	MDA-MB-231	Silenced RhoA in TNBC, and reduced TNBC tumor without toxicity	[113]
CDK1 and c-Myc	PEG–PLA NPs	SUM149 and BT549	Inhibition of CDK1 expression in cMyc overexpressed TNBC reduced cell viability through apoptotic cell death demonstrating synthetic lethality between cMyc with CDK1 in TNBC cells	[142]
Survivin	Lipid-substituted polymer NPs	MDA-MB-231	Decreased cancer cell viability, down-regulated survivin protein, inhibited tumor cell growth reduced chemoresistance	[93]
Survivin	PEG_2K_–PE–PM	MDA-MB-231	Reduced survivin expression in resistant cancer cells, triggered microtubule destabilization and significantly inhibited TNBC tumor growth	[93]
Survivin	DLP/siRNA	MDA-MB-231	Induced apoptotic cell death of TNBC cell lines effectively, suppressed cancer cell stemness, and inhibited tumor development	[95]
TWIST	PAMAM–dendrimer NPs	SUM1315	Decreased TWIST expression along with phenotypic variations, and reduced cancer cell migration, inhibited TWIST-inspired amplification of mesenchymal marker, preventing TNBC tumor cells	[111,112]
Lipocalin 2	ICAM-1 conjugated Liposomes NPs	MDA-MB-231	Demonstrated noteworthy inhibition of VEGF from MDA-MB-231 cells, resulting in diminished angiogenesis both in vitro and in vivo	[114]
EGFR	CPP-loaded Nanobubbles	MDA-MB-231	Exhibited significant downregulation of EGFR mRNA and protein in a xenografted tumor model of TNBC cells; suppressed miRNA-21, proliferation of TNBC cells via controlling EGFR and subsequently inhibiting the PI3K/AKT and ERK1/2 signaling axis	[10,126,127,128,129]
DANCR	RGD–PEG–ECO NPs	MDA-MB-231	Reduced EDB-FN expression, demonstrated effectiveness against both TNBC and MCF-7 cell lines	[138,139]
MDR1	Layer by layer NPs (depositing alternately siRNA and poly-L-arginine on NPs)	MDA-MB-468	Inhibited MDR1 protein, increased doxorubicin sensitivity 4-fold and significantly decreased tumor volume	[91]
MDR1	siRNA–NPs with PLL and hyaluronic acid	MDA-MB-231	Downregulated ABCB1 and ABCG2 and increased the sensitivity of TNBC cells to doxorubicin and paclitaxel	[90,91]
mTORC2	Silicon NPs	BT474, MDA-MB-361, MDA-MB-231, SKBR3	Demonstrated selective mTOR2 inhibition in TNBC, decreased Akt phosphorylation, and tumor growth in TNBC	[145]
Rictor (mTORC2)	50B8–DP100 siNP	MDA-MB-231	Significantly inhibited mTOR2 activity in TNBC, potentially reduced Rictor expression in mTOR/PI3K active TNBC tumors	[134,135]
ATM protein	Nanoliposome	MDA-MB-231, SKBR3	Reduced ATM protein expression in TNBC, stimulated cytotoxic T lymphocytes and controlled the immune-responsive tumor microenvironment triggering the cGAS-STING pathway	[137]
STAT3	Cholesterol–siRNA and cationic PLL [30]-PEG_5K_	4T1	Downregulated STAT3, suppressed mRNA in 4T1 cells	[115]
TRPC6	Shamporter coupled with siTRPC6	MDA-MB-231	Substantially silenced TRPC6 protein level, inhibited TNBC growth	[125]
MCL1	siRNA–lipid–albumin conjugates	MDA-MB-231	Silenced MCL1 expression and cMyc expression, reduced cancer cell stemness, and inhibited TNBC cell growth	[105,106]

## Data Availability

Not applicable.

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
