# Peer review of "Advances in siRNA Drug Delivery Strategies for Targeted TNBC Therapy"

_bioengineering, 2024, doi:10.3390/bioengineering11080830_

Round 1

Reviewer 1 Report (Previous Reviewer 1)

Comments and Suggestions for Authors

This review-based manuscript discusses the challenges of treating triple-negative breast cancer (TNBC) due to its lack of targeted therapy options. It emphasizes the potential of siRNA-based therapies to address this issue by silencing specific genes involved in TNBC tumor growth and progression. This review explores recent advances in nanomaterial- and nanobiomaterial-based siRNA delivery platforms, highlighting their ability to overcome delivery barriers and enhance the therapeutic efficacy of siRNA in TNBC. The authors also include various therapeutic targets for siRNA in TNBC, including genes involved in drug resistance, apoptosis, cell cycle regulation, and angiogenesis.

 However, the quality of this manuscript could be significantly improved by addessing the major concerns listed below:

Major Concerns:

1.      The manuscript mentions various targets for siRNA in TNBC but does not provide sufficient detail on specific genes or pathways. For instance, targeting the PI3K/AKT pathway is mentioned, but detailed mechanisms or specific siRNA sequences used to target this pathway are not discussed. More detailed mechanistic insights into how siRNA affects specific genes and pathways in TNBC would enhance the quality of this manuscript.

2.      While preclinical studies are mentioned, there is limited discussion on the translation of these findings to clinical settings. The manuscript should address the challenges of moving from preclinical models to clinical trials, including issues such as delivery efficiency, off-target effects, and patient variability. If there is any case studies or most recent research involving siRNA therapies for TNBC? If yes, I would like to suggest including them.

3.      The connection between sections is sort of not smooth (logical disconnected somehow). For example, the section on biomarkers and TILs could be better integrated into the discussion of therapeutic targets and delivery strategies. A more seamless integration of these topics would improve the overall coherence of the manuscript.

4.      Several statements in the manuscript lack proper citations, making it difficult to verify the information presented. For example, specific data points such as global cancer statistics and details about siRNA mechanisms need appropriate references.

5.      The manuscript implies the potential of siRNA therapies but does not explicitly state why siRNA is a promising approach for TNBC. A clear rationale, supported by specific examples of siRNA success in TNBC models, would strengthen the manuscript's argument.

Comments on the Quality of English Language

I think moderate editing of the English language is required to improve readability. Simplify complex sentences and correct grammatical errors to enhance clarity.   For example, "The most aggressive, highly heterogeneous subtype of breast cancer is TNBC, which lacks expression of ER, PR, and HER2 markers." could be revised into "TNBC is the most aggressive and highly heterogeneous subtype of breast cancer, lacking the expression of ER, PR, and HER2 markers.". 

In addition, there are some vague language issues such as "Several studies have shown that this approach is effective.". "This approach" should be specific. 

Author Response

Dear Editor

Thank you for sending us review report. We have now revised our manuscript according valuable comments and suggestions from the reviewers. Hopefully, you will find the current version as suitable for publication in Bioengineering. We thank reviewers for comments and suggestions. We have replied each comments point-by-point.

Please find below the responses to reviewer.

Reviewer 1

This review-based manuscript discusses the challenges of treating triple-negative breast cancer (TNBC) due to its lack of targeted therapy options. It emphasizes the potential of siRNA-based therapies to address this issue by silencing specific genes involved in TNBC tumor growth and progression. This review explores recent advances in nanomaterial- and nanobiomaterial-based siRNA delivery platforms, highlighting their ability to overcome delivery barriers and enhance the therapeutic efficacy of siRNA in TNBC. The authors also include various therapeutic targets for siRNA in TNBC, including genes involved in drug resistance, apoptosis, cell cycle regulation, and angiogenesis.

 However, the quality of this manuscript could be significantly improved by addessing the major concerns listed below:

Ans. Thank you for your comments.

Major Concerns:

1.The manuscript mentions various targets for siRNA in TNBC but does not provide sufficient detail on specific genes or pathways. For instance, targeting the PI3K/AKT pathway is mentioned, but detailed mechanisms or specific siRNA sequences used to target this pathway are not discussed. More detailed mechanistic insights into how siRNA affects specific genes and pathways in TNBC would enhance the quality of this manuscript.

Ans. We described in the text.

  1. While preclinical studies are mentioned, there is limited discussion on the translation of these findings to clinical settings. The manuscript should address the challenges of moving from preclinical models to clinical trials, including issues such as delivery efficiency, off-target effects, and patient variability. If there is any case studies or most recent research involving siRNA therapies for TNBC? If yes, I would like to suggest including them.

Ans. TNBC therapies are still growing mostly in pre-clinical stages. To the best of our knowledge no clinical trials based on TNBC or still not reported. There are some clinical trials for solid tumors that may be helpful for TNBC.

  1. The connection between sections is sort of not smooth (logical disconnected somehow). For example, the section on biomarkers and TILs could be better integrated into the discussion of therapeutic targets and delivery strategies. A more seamless integration of these topics would improve the overall coherence of the manuscript.

Ans. We tried to improve.

  1. Several statements in the manuscript lack proper citations, making it difficult to verify the information presented. For example, specific data points such as global cancer statistics and details about siRNA mechanisms need appropriate references.

Ans. Corrected. Some references added.

  1. The manuscript implies the potential of siRNA therapies but does not explicitly state why siRNA is a promising approach for TNBC. A clear rationale, supported by specific examples of siRNA success in TNBC models, would strengthen the manuscript's argument.

Ans. Added some comments and references in the text.

Comments on the Quality of English Language

I think moderate editing of the English language is required to improve readability. Simplify complex sentences and correct grammatical errors to enhance clarity.   For example, "The most aggressive, highly heterogeneous subtype of breast cancer is TNBC, which lacks expression of ER, PR, and HER2 markers." could be revised into "TNBC is the most aggressive and highly heterogeneous subtype of breast cancer, lacking the expression of ER, PR, and HER2 markers.". 

In addition, there are some vague language issues such as "Several studies have shown that this approach is effective.". "This approach" should be specific. 

Ans. Thank you.

comments and Suggestions for Authors

Reviewer 2

The manuscript “Advances with siRNA Drug Delivery Strategies for Targeted TNBC Therapy”, revised and presented by Md Abdus Subhan, and Vladimir P. Torchilin, has changed significantly compared to the original version. The authors took into account the reviewers’ comments; large inserts appeared in the text to improve the review. The review provides comprehensive insight into the development and use of drugs for the treatment of triple negative breast cancer. The tables included in the text will be useful as reference material for researchers. The authors presented recent publications and outlined their results. Unfortunately, not all results have been analyzed critically, but the review can be useful as a collection of facts, which is sometimes needed in scientific work.

The level of the manuscript has improved and formally it meets the requirements for publication.

Ans. Thank you.

Let me express an “informal” opinion; perhaps it will be useful to authors in the future.

The text gives the impression of “Morse code” - individual fragments do not flow into each other. It seems that everything is written correctly, but you read the text as if you were hardly moving from stone to stone. Obviously, the style of the text is a feature of the authors, however, ease of reading ensures the correct perception of the meaning. I can’t demand “nice” presentation from the authors, but you need to carefully proofread the text - there are still errors and typos.

So, the review "Advances with siRNA Drug Delivery Strategies for Targeted TNBC Therapy", may be published after typing and grammatical errors are corrected.

Ans. Thank you. Corrected.

Comments on the Quality of English Language

Please check the grammar carefully - there are errors

Reviewer 3

Comments and Suggestions for Authors

The author reviewed the recent advances in siRNA drug delivery strategies for targeted triple-negative breast cancer therapy. The manuscript was well-written and supported by sufficient references. The manuscript discussed the significant roles of biomarkers such as genes, and proteins. However, the manuscript needs some minor corrections as below.

Ans. Thank you.

  • All adopted figures need to be enhanced in resolution.

Ans. Improved

  • Legend of figure 2 need more clarification. Which boxes are the target and which boxes are the inhibitors?

Ans. Corrected.

  • The legend of figure 3 is confusing. We suggest the revised version: "Mechanism of RNA interference in siRNA therapeutics for gene silencing.”

Ans. Corrected.

  • Figure 4 is distorted. Please obtain and reproduce a better-quality figure. Also, the legend lacks details.

Ans. Replaced and modified.

  • Line 352, there should be a space between “FOXM1siRNA”, “FOXM1 siRNA”.

Ans. Corrected.

  • Figure 5 legend needs to be more detailed.

Ans. Modified

  • Figure 5 was repeat labeled in Figure 6.

Ans. Corrected

  • Please check all reference formats. Some of the references were cut in the middle.  

Ans. Corrected.

Reviewer 2 Report (Previous Reviewer 2)

Comments and Suggestions for Authors

The manuscript “Advances with siRNA Drug Delivery Strategies for Targeted TNBC Therapy”, revised and presented by Md Abdus Subhan, and Vladimir P. Torchilin, has changed significantly compared to the original version. The authors took into account the reviewers’ comments; large inserts appeared in the text to improve the review. The review provides comprehensive insight into the development and use of drugs for the treatment of triple negative breast cancer. The tables included in the text will be useful as reference material for researchers. The authors presented recent publications and outlined their results. Unfortunately, not all results have been analyzed critically, but the review can be useful as a collection of facts, which is sometimes needed in scientific work.

The level of the manuscript has improved and formally it meets the requirements for publication.

Let me express an “informal” opinion; perhaps it will be useful to authors in the future.

The text gives the impression of “Morse code” - individual fragments do not flow into each other. It seems that everything is written correctly, but you read the text as if you were hardly moving from stone to stone. Obviously, the style of the text is a feature of the authors, however, ease of reading ensures the correct perception of the meaning. I can’t demand “nice” presentation from the authors, but you need to carefully proofread the text - there are still errors and typos.

So, the review "Advances with siRNA Drug Delivery Strategies for Targeted TNBC Therapy", may be published after typing and grammatical errors are corrected.

Comments on the Quality of English Language

Please check the grammar carefully - there are errors

Author Response

Dear Editor

Thank you for sending us review report. We have now revised our manuscript according valuable comments and suggestions from the reviewers. Hopefully, you will find the current version as suitable for publication in Bioengineering. We thank reviewers for comments and suggestions. We have replied each comments point-by-point.

Please find below the responses to reviewer.

Reviewer 1

This review-based manuscript discusses the challenges of treating triple-negative breast cancer (TNBC) due to its lack of targeted therapy options. It emphasizes the potential of siRNA-based therapies to address this issue by silencing specific genes involved in TNBC tumor growth and progression. This review explores recent advances in nanomaterial- and nanobiomaterial-based siRNA delivery platforms, highlighting their ability to overcome delivery barriers and enhance the therapeutic efficacy of siRNA in TNBC. The authors also include various therapeutic targets for siRNA in TNBC, including genes involved in drug resistance, apoptosis, cell cycle regulation, and angiogenesis.

 However, the quality of this manuscript could be significantly improved by addessing the major concerns listed below:

Ans. Thank you for your comments.

Major Concerns:

1.The manuscript mentions various targets for siRNA in TNBC but does not provide sufficient detail on specific genes or pathways. For instance, targeting the PI3K/AKT pathway is mentioned, but detailed mechanisms or specific siRNA sequences used to target this pathway are not discussed. More detailed mechanistic insights into how siRNA affects specific genes and pathways in TNBC would enhance the quality of this manuscript.

Ans. We described in the text.

  1. While preclinical studies are mentioned, there is limited discussion on the translation of these findings to clinical settings. The manuscript should address the challenges of moving from preclinical models to clinical trials, including issues such as delivery efficiency, off-target effects, and patient variability. If there is any case studies or most recent research involving siRNA therapies for TNBC? If yes, I would like to suggest including them.

Ans. TNBC therapies are still growing mostly in pre-clinical stages. To the best of our knowledge no clinical trials based on TNBC or still not reported. There are some clinical trials for solid tumors that may be helpful for TNBC.

  1. The connection between sections is sort of not smooth (logical disconnected somehow). For example, the section on biomarkers and TILs could be better integrated into the discussion of therapeutic targets and delivery strategies. A more seamless integration of these topics would improve the overall coherence of the manuscript.

Ans. We tried to improve.

  1. Several statements in the manuscript lack proper citations, making it difficult to verify the information presented. For example, specific data points such as global cancer statistics and details about siRNA mechanisms need appropriate references.

Ans. Corrected. Some references added.

  1. The manuscript implies the potential of siRNA therapies but does not explicitly state why siRNA is a promising approach for TNBC. A clear rationale, supported by specific examples of siRNA success in TNBC models, would strengthen the manuscript's argument.

Ans. Added some comments and references in the text.

Comments on the Quality of English Language

I think moderate editing of the English language is required to improve readability. Simplify complex sentences and correct grammatical errors to enhance clarity.   For example, "The most aggressive, highly heterogeneous subtype of breast cancer is TNBC, which lacks expression of ER, PR, and HER2 markers." could be revised into "TNBC is the most aggressive and highly heterogeneous subtype of breast cancer, lacking the expression of ER, PR, and HER2 markers.". 

In addition, there are some vague language issues such as "Several studies have shown that this approach is effective.". "This approach" should be specific. 

Ans. Thank you.

comments and Suggestions for Authors

Reviewer 2

The manuscript “Advances with siRNA Drug Delivery Strategies for Targeted TNBC Therapy”, revised and presented by Md Abdus Subhan, and Vladimir P. Torchilin, has changed significantly compared to the original version. The authors took into account the reviewers’ comments; large inserts appeared in the text to improve the review. The review provides comprehensive insight into the development and use of drugs for the treatment of triple negative breast cancer. The tables included in the text will be useful as reference material for researchers. The authors presented recent publications and outlined their results. Unfortunately, not all results have been analyzed critically, but the review can be useful as a collection of facts, which is sometimes needed in scientific work.

The level of the manuscript has improved and formally it meets the requirements for publication.

Ans. Thank you.

Let me express an “informal” opinion; perhaps it will be useful to authors in the future.

The text gives the impression of “Morse code” - individual fragments do not flow into each other. It seems that everything is written correctly, but you read the text as if you were hardly moving from stone to stone. Obviously, the style of the text is a feature of the authors, however, ease of reading ensures the correct perception of the meaning. I can’t demand “nice” presentation from the authors, but you need to carefully proofread the text - there are still errors and typos.

So, the review "Advances with siRNA Drug Delivery Strategies for Targeted TNBC Therapy", may be published after typing and grammatical errors are corrected.

Ans. Thank you. Corrected.

Comments on the Quality of English Language

Please check the grammar carefully - there are errors

Reviewer 3

Comments and Suggestions for Authors

The author reviewed the recent advances in siRNA drug delivery strategies for targeted triple-negative breast cancer therapy. The manuscript was well-written and supported by sufficient references. The manuscript discussed the significant roles of biomarkers such as genes, and proteins. However, the manuscript needs some minor corrections as below.

Ans. Thank you.

  • All adopted figures need to be enhanced in resolution.

Ans. Improved

  • Legend of figure 2 need more clarification. Which boxes are the target and which boxes are the inhibitors?

Ans. Corrected.

  • The legend of figure 3 is confusing. We suggest the revised version: "Mechanism of RNA interference in siRNA therapeutics for gene silencing.”

Ans. Corrected.

  • Figure 4 is distorted. Please obtain and reproduce a better-quality figure. Also, the legend lacks details.

Ans. Replaced and modified.

  • Line 352, there should be a space between “FOXM1siRNA”, “FOXM1 siRNA”.

Ans. Corrected.

  • Figure 5 legend needs to be more detailed.

Ans. Modified

  • Figure 5 was repeat labeled in Figure 6.

Ans. Corrected

  • Please check all reference formats. Some of the references were cut in the middle.  

Ans. Corrected.

Reviewer 3 Report (Previous Reviewer 3)

Comments and Suggestions for Authors

The author reviewed the recent advances in siRNA drug delivery strategies for targeted triple-negative breast cancer therapy. The manuscript was well-written and supported by sufficient references. The manuscript discussed the significant roles of biomarkers such as genes, and proteins. However, the manuscript needs some minor corrections as below.

1)     All adopted figures need to be enhanced in resolution.

2)     Legend of figure 2 need more clarification. Which boxes are the target and which boxes are the inhibitors?

3)     The legend of figure 3 is confusing. We suggest the revised version: "Mechanism of RNA interference in siRNA therapeutics for gene silencing.”

4)     Figure 4 is distorted. Please obtain and reproduce a better-quality figure. Also, the legend lacks details.

5)     Line 352, there should be a space between “FOXM1siRNA”, “FOXM1 siRNA”.

6)     Figure 5 legend needs to be more detailed.

7)     Figure 5 was repeat labeled in Figure 6.

8)     Please check all reference formats. Some of the references were cut in the middle.  

Author Response

Dear Editor

Thank you for sending us review report. We have now revised our manuscript according valuable comments and suggestions from the reviewers. Hopefully, you will find the current version as suitable for publication in Bioengineering. We thank reviewers for comments and suggestions. We have replied each comments point-by-point.

Please find below the responses to reviewer.

Reviewer 1

This review-based manuscript discusses the challenges of treating triple-negative breast cancer (TNBC) due to its lack of targeted therapy options. It emphasizes the potential of siRNA-based therapies to address this issue by silencing specific genes involved in TNBC tumor growth and progression. This review explores recent advances in nanomaterial- and nanobiomaterial-based siRNA delivery platforms, highlighting their ability to overcome delivery barriers and enhance the therapeutic efficacy of siRNA in TNBC. The authors also include various therapeutic targets for siRNA in TNBC, including genes involved in drug resistance, apoptosis, cell cycle regulation, and angiogenesis.

 However, the quality of this manuscript could be significantly improved by addessing the major concerns listed below:

Ans. Thank you for your comments.

Major Concerns:

1.The manuscript mentions various targets for siRNA in TNBC but does not provide sufficient detail on specific genes or pathways. For instance, targeting the PI3K/AKT pathway is mentioned, but detailed mechanisms or specific siRNA sequences used to target this pathway are not discussed. More detailed mechanistic insights into how siRNA affects specific genes and pathways in TNBC would enhance the quality of this manuscript.

Ans. We described in the text.

  1. While preclinical studies are mentioned, there is limited discussion on the translation of these findings to clinical settings. The manuscript should address the challenges of moving from preclinical models to clinical trials, including issues such as delivery efficiency, off-target effects, and patient variability. If there is any case studies or most recent research involving siRNA therapies for TNBC? If yes, I would like to suggest including them.

Ans. TNBC therapies are still growing mostly in pre-clinical stages. To the best of our knowledge no clinical trials based on TNBC or still not reported. There are some clinical trials for solid tumors that may be helpful for TNBC.

  1. The connection between sections is sort of not smooth (logical disconnected somehow). For example, the section on biomarkers and TILs could be better integrated into the discussion of therapeutic targets and delivery strategies. A more seamless integration of these topics would improve the overall coherence of the manuscript.

Ans. We tried to improve.

  1. Several statements in the manuscript lack proper citations, making it difficult to verify the information presented. For example, specific data points such as global cancer statistics and details about siRNA mechanisms need appropriate references.

Ans. Corrected. Some references added.

  1. The manuscript implies the potential of siRNA therapies but does not explicitly state why siRNA is a promising approach for TNBC. A clear rationale, supported by specific examples of siRNA success in TNBC models, would strengthen the manuscript's argument.

Ans. Added some comments and references in the text.

Comments on the Quality of English Language

I think moderate editing of the English language is required to improve readability. Simplify complex sentences and correct grammatical errors to enhance clarity.   For example, "The most aggressive, highly heterogeneous subtype of breast cancer is TNBC, which lacks expression of ER, PR, and HER2 markers." could be revised into "TNBC is the most aggressive and highly heterogeneous subtype of breast cancer, lacking the expression of ER, PR, and HER2 markers.". 

In addition, there are some vague language issues such as "Several studies have shown that this approach is effective.". "This approach" should be specific. 

Ans. Thank you.

comments and Suggestions for Authors

Reviewer 2

The manuscript “Advances with siRNA Drug Delivery Strategies for Targeted TNBC Therapy”, revised and presented by Md Abdus Subhan, and Vladimir P. Torchilin, has changed significantly compared to the original version. The authors took into account the reviewers’ comments; large inserts appeared in the text to improve the review. The review provides comprehensive insight into the development and use of drugs for the treatment of triple negative breast cancer. The tables included in the text will be useful as reference material for researchers. The authors presented recent publications and outlined their results. Unfortunately, not all results have been analyzed critically, but the review can be useful as a collection of facts, which is sometimes needed in scientific work.

The level of the manuscript has improved and formally it meets the requirements for publication.

Ans. Thank you.

Let me express an “informal” opinion; perhaps it will be useful to authors in the future.

The text gives the impression of “Morse code” - individual fragments do not flow into each other. It seems that everything is written correctly, but you read the text as if you were hardly moving from stone to stone. Obviously, the style of the text is a feature of the authors, however, ease of reading ensures the correct perception of the meaning. I can’t demand “nice” presentation from the authors, but you need to carefully proofread the text - there are still errors and typos.

So, the review "Advances with siRNA Drug Delivery Strategies for Targeted TNBC Therapy", may be published after typing and grammatical errors are corrected.

Ans. Thank you. Corrected.

Comments on the Quality of English Language

Please check the grammar carefully - there are errors

Reviewer 3

Comments and Suggestions for Authors

The author reviewed the recent advances in siRNA drug delivery strategies for targeted triple-negative breast cancer therapy. The manuscript was well-written and supported by sufficient references. The manuscript discussed the significant roles of biomarkers such as genes, and proteins. However, the manuscript needs some minor corrections as below.

Ans. Thank you.

  • All adopted figures need to be enhanced in resolution.

Ans. Improved

  • Legend of figure 2 need more clarification. Which boxes are the target and which boxes are the inhibitors?

Ans. Corrected.

  • The legend of figure 3 is confusing. We suggest the revised version: "Mechanism of RNA interference in siRNA therapeutics for gene silencing.”

Ans. Corrected.

  • Figure 4 is distorted. Please obtain and reproduce a better-quality figure. Also, the legend lacks details.

Ans. Replaced and modified.

  • Line 352, there should be a space between “FOXM1siRNA”, “FOXM1 siRNA”.

Ans. Corrected.

  • Figure 5 legend needs to be more detailed.

Ans. Modified

  • Figure 5 was repeat labeled in Figure 6.

Ans. Corrected

  • Please check all reference formats. Some of the references were cut in the middle.  

Ans. Corrected.

Round 2

Reviewer 1 Report (Previous Reviewer 1)

Comments and Suggestions for Authors

Based on the revisions made in this updated manuscript, the authors have substantially addressed the major concerns raised in my review comments. The revised manuscript now includes detailed mechanistic insights, expanded discussion on preclinical to clinical translation, improved logical flow, additional citations, and a clear rationale for the use of siRNA in TNBC therapy.

This manuscript is a resubmission of an earlier submission. The following is a list of the peer review reports and author responses from that submission.

Round 1

Reviewer 1 Report

Comments and Suggestions for Authors

Summary:

This review discusses the challenges of treating triple-negative breast cancer (TNBC) and the potential of siRNA-based therapies as a promising treatment approach. The authors provide an overview of TNBC, its biomarkers, and current treatment strategies. They then delve into the mechanisms of siRNA therapy and highlight recent advances in siRNA delivery platforms, particularly those utilizing nanomaterials and nanobiomaterials. The review concludes by emphasizing the potential of siRNA-based therapies to improve TNBC treatment outcomes.

There are some major concerns needed to be addressed:

Major Concerns:

1.        Since the title of this manuscript is “siRNA Drug Delivery Strategies for Targeted TNBC Therapy”, thus I think the author should spend more  content in the Introduction section about the current “siRNA therapy” and “ targeted TNBC” instead of over drafting about the nanoparticles, especially nanobiomaterials.

2.        The manuscript lacks a clear and logical flow. The sections are not well-organized, and the transitions between topics are abrupt. For example, the discussion of biomarkers and TILs in section 2 seems disconnected from the subsequent sections on therapeutic targets and siRNA therapies.

3.        The review covers a wide range of topics but lacks depth in some areas. For instance, the discussion of current therapeutic targets in section 3 is superficial and does not provide a critical analysis of their advantages and limitations.

4.        While the authors cite several studies, the literature review could be more comprehensive and up to date. Some recent advances in siRNA delivery platforms and clinical trials are not mentioned.

5.        The review lacks a critical analysis of the current state of the field. The authors present the information but do not offer their own insights or perspectives on the challenges and opportunities in siRNA-based TNBC therapy.

6.        The manuscript requires significant editing for grammar, language, and clarity. There are numerous grammatical errors, awkward sentence structures, and typos throughout the text.

Comments on the Quality of English Language

1. There are several language issues, for example:"Chemotherapy, radiation therapy, and surgery is the common therapies for TNBC."  The verb should be "are."

2. There are some issues with the article usage as well. For example""The initial treatment of TNBC is surgery and radiation therapy followed by chemotherapy in a conventional method." It should be "followed by chemotherapy using a conventional method."

Author Response

Dear Reviewer

Thank you for your comments and suggestion. Please see the detail in the attached file.

Best regards

Md Abdus Subhan

Reviewer 2 Report

Comments and Suggestions for Authors

The title “Advances with siRNA Drug Delivery Strategies for Targeted TNBC Therapy” of the manuscript suggests an analysis of publications on advances in breast cancer therapy using targeted siRNA delivery. In the Introduction, the authors note that they will first describe current therapeutic strategies before focusing on nanomaterial-based siRNA delivery systems and effective targeted therapy approaches to combat TNBC.

The authors describe in detail biomarkers, T-lymphocytes, genes, and their role in development of TNBC.

Interesting data are presented concerning genes related to TNBC development and breast cancer risk factors. The Tables 1 and 2 could be useful for specialists, however, references to relevant publications in Table 2 are required. The section “Currently utilized therapeutic targets in TNBC” provides brief, but comprehensive overview of the topic. 

The section on siRNA therapy presents a description of the mechanisms of RNA interference and their involvement in various cellular processes. Finally, the authors got to the subject declared in the title of the article: “Targeted siRNA therapy for TNBC” (section 5).

This topic is undoubtedly very complicated and requires detailed and critical analysis. I am confused by the optimistic attitude of the authors regarding the results of using siRNA specifically delivered to TNBC cancer cells. The authors analyze in detail all possible molecular targets for microRNAs, but more than half of the articles were published about 10 or more years ago. Confirming later publications are not presented, which may indicate the revealed futility of further research in relation to a specific experimental system. This is alarming: as a rule, really promising research continues and develops, with drugs undergoing preclinical and clinical studies. In the section authors describe numerous molecular targets for siRNA therapy and analyze the difficulties in developing this approach.

Table 3. Anticancer effect-mediated by siRNA drug-based TNBC therapy in preclinical settings: Please add exactly what effects are noted in each article.

 The final paragraph of the section states:

(Line 627)  siRNA therapeutics for enhanced therapy of TNBC patients are effective and promising. Several siRNA-induced NPs have either started phase I, phase II, and phase III clinical trials or have been approved by the FDA for therapy of various types of cancers including solid tumors that may be applied for TNBC tumors [73, 153]. Recently, four siRNA drugs have received approval from the FDA; patisiran in 2018, givosiran in 2019, lumasiran in 2020 and inclisiran in 2021 [117]. As a result, siRNA drugs have lately received enhanced consideration. Fabrication of an appropriate siRNA-induced delivery platform is crucial for cancer therapy including TNBC in clinical setting. These strategies should help impede TNBC and improve the survival of the patients.

 I cannot agree that the publications cited in this section of the article confirm the effectiveness of siRNA therapy in cases of TNBC. References 73 and 153 are reviews, devoted to general considerations about siRNA applications. I was very pleased when siRNA drugs were approved by FDA for clinical use. However, if you cite them as an example of improving the situation with TNBC, please provide relevant publications.

It is well known that different types of cancer require different approaches to treatment; this has long been an axiom. It is also known that results in cell cultures are often not reproduced even in animal tumor models. Fact: a very small proportion of synthesized drugs reach clinical trials. The content of the manuscript does not correspond to its title “Advances with siRNA Drug Delivery Strategies for Targeted TNBC Therapy”, which declares success in the treatment of TNBC due to siRNA therapy. I am not an opponent of the use of siRNA; however, I believe that the review should reflect the real situation. I am not an opponent of nanomaterials, and I believe that little attention is paid to them in this review.

Overall, the authors have done a great job of analyzing the molecular characteristics of TNBC using current data and it is worthy of publication, but I believe that the siRNA portion of the manuscript should be revised in accordance with my comments. I also recommend that authors think carefully about the title of the work and try to compress the descriptions; the text looks “blurry”.

Comments on the Quality of English Language

The article as a whole is written in understandable language, however, I recommend showing the text to a native English speaker.

Author Response

Dear Reviewer

Thank you for your comments valuable comments. Please see the detail in the attached file.

Best regards

Md Abdus Subhan

Reviewer 3 Report

Comments and Suggestions for Authors

The author reviewed the recent advances of siRNA drug delivery strategies for targeted triple negative breast cancer therapy. The review covered sufficient topics for discussion with recent references. However, the authors should improve on those below points.

1)     Please improve the quality of reproduction materials. The figures in this review manuscript are poorly obtained with low resolution.

2)     Please provide the references for each gene in table 1 and also references for each biomarker in table 2 is missing. Table 1 and 2 should follow the format of table 3.

3)     Table 2 is merely a crop-out from a table in reference 43. Please put the copyright for the legend of the table. Likewise for table 1.

4)     Section 5 which is the main topic for this review is too long and hard to follow. Please divide the section into sub-sections with headings for the reader’s ease of following.

5)     The reference format is not consistent. Some names and titles of references were underlined unnecessarily.

6)     The paragraph format for reference is not consistent. A lot of references are broken into many different lines.

7)     Section 1,2,3,4 provides an overview of background mechanism, and current methods in TNBC treatments. Those are too long and disproportionate to the main section 5. Those sections have a high percentage of similarity to the author’s previous publication. We suggest the author summarize those sections to provide more focus on section 5. 

Author Response

Dear Reviewer

Thank you for your valuable comments and suggestions. Please see the detail in the attached file.

Best regards

Md Abdus Subhan
